# Impact of Nisin and Nisin-Producing *Lactococcus lactis* ssp. *lactis* on *Clostridium tyrobutyricum* and Bacterial Ecosystem of Cheese Matrices

**DOI:** 10.3390/foods10040898

**Published:** 2021-04-19

**Authors:** Hebatoallah Hassan, Daniel St-Gelais, Ahmed Gomaa, Ismail Fliss

**Affiliations:** 1STELA Dairy Research Center, Institute of Nutrition and Functional Foods, Université Laval, Québec, QC G1V 0A6, Canada; ismail.fliss@fsaa.ulaval.ca; 2Institute of Graduate Studies and Research, Alexandria University, Alexandria 21526, Egypt; 3Food Research and Development Centre, Agriculture and Agri-Food Canada, Saint-Hyacinthe, QC J2S 8E3, Canada; damilk83@hotmail.com; 4National Research Center, Nutrition and Food Science Department, Cairo 12622, Egypt; ahmed.el-sayed@mail.mcgill.ca

**Keywords:** Nisin, protective starter, antimicrobial peptides, Cheddar cheese slurry, *Clostridium tyrobutyricum*, *Lactococcus lactis*, dairy products

## Abstract

*Clostridium tyrobutyricum* spores survive milk pasteurization and cause late blowing of cheeses and significant economic loss. The effectiveness of nisin-producing *Lactococcus lactis* ssp. *lactis* 32 as a protective strain for control the *C. tyrobutyricum* growth in Cheddar cheese slurry was compared to that of encapsulated nisin-A. The encapsulated nisin was more effective, with 1.0 log_10_ reductions of viable spores after one week at 30 °C and 4 °C. Spores were not detected for three weeks at 4 °C in cheese slurry made with 1.3% salt, or during week 2 with 2% salt. Gas production was observed after one week at 30 °C only in the control slurry made with 1.3% salt. In slurry made with the protective strain, the reduction in *C. tyrobutyricum* count was 0.6 log_10_ in the second week at 4 °C with both salt concentration. At 4 °C, nisin production started in week 2 and reached 97 µg/g after four weeks. Metabarcoding analysis targeting the sequencing of 16S rRNA revealed that the genus *Lactococcus* dominated for four weeks at 4 °C. In cheese slurry made with 2% salt, the relative abundance of the genus *Clostridium* decreased significantly in the presence of nisin or the protective strain. The results indicated that both strategies are able to control the growth of *Clostridium* development in Cheddar cheese slurries.

## 1. Introduction

*Clostridium (C.) tyrobutyricum* is a Gram-positive, obligately anaerobic endospore-forming bacterium. In semi-hard and hard cheeses, it is capable of metabolizing lactic acid and producing butyric acid, hydrogen and carbon dioxide, which leads to major defects such as late blowing, swelling and splitting of blocks, brittle texture and foul odor [1,2]. Since the cost of bringing cheese to this stage of ripening is considerable, these defects cause significant economic losses [3,4]. Different approaches have been used to eliminate *C. tyrobutricum* from value-added food products, especially cheese. These include physical methods like bactofugation or microfiltration of milk. Such methods remain expensive in addition to causing significant modification of milk composition. Moreover, microfiltration should be applied to skim milk since fat globules clog filters [5,6]. Antimicrobial substances such as nitrate, lysozyme or salt have been used to prevent growth of *C. tyrobutyricum* [7,8]. The problem with nitrate is its conversion to nitrite by milk bacteria that express xanthine oxidase [9], possibly leading to the formation of nitrosamine, a carcinogen [7]. Also, such methods are not permitted for the Protected Designation of Origin (PDO) products [10]. The problem with lysozyme is that it comes from hen’s egg, to which some individuals are allergic [11]. Salt is effective at concentrations of 0.5–2.5% in soft, hard and semi hard cheeses and 3.5–5% in blue cheese [12]. However, salty foods have fallen into disfavor because their consumption increases the risk of hypertension, cardiovascular disease and kidney failure [13]. Salt is an important ingredient in Cheddar cheese, playing a major role in the development of flavor, texture and microbiological stability [14]. Reducing salt concentration has become a major challenge to producers of processed foods in general, including cheese [12]. Therefore, new ways of inhibiting *C. tyrobutyricum* in cheese are needed.

Promising alternative food preservatives are protective bacterial cultures or concentrated active metabolites from lactic acid bacteria (LAB). They are known to produce a wide variety of antimicrobial substances, such as bacteriocins [15,16]. Bacteriocins are antimicrobial peptides or proteins, sometimes of complex structure, synthesized by ribosomes and secreted with or without post-translational modification [17].

Although many bacteriocins have been discovered, nisin remains the only one approved as a food additive by the European Food Safety Authority (EFSA) and Food and Drug Administration (FDA) and has number E234 [18]. Consisting of 34 amino acid residues, nisin is mainly produced by *Lactococcus lactis* strains. Many types are known, which differ in some amino acids like nisin A, Z, F and U [19]. Nisin has an antimicrobial effect against a wide range of Gram-positive foodborne and spoilage bacteria [20,21].

Nisin was discovered in 1928, but its efficacy in food matrices is based on limited study [19]. Food-grade nisin is difficult to produce on a large scale. The peptide seems to be unstable in food matrices, especially in foods fermented with LAB, which are known to produce many proteases. Its hydrophobicity causes it to bind strongly to certain food constituents which decrease its effectiveness during product storage. For these reasons, the use of nisin as a food additive has been scant [22,23]. The aim of the present study was to evaluate the effectiveness of a nisin-producing *Lactococcus lactis* strain as an inhibitor of the growth and development of *C. tyrobutyricum* in Cheddar cheese and to compare it with encapsulated nisin.

## 2. Materials and Methods

### 2.1. Bacterial Cultures

*Lactococcus lactis* ssp. *lactis* CUC-H and *Lactococcus lactis* ssp. *cremoris* CUC 222 were obtained from Agriculture and Agri-Food Canada, St-Hyacinthe Research and Development Center (used as a starter). The nisin-A producing *Lactococcus lactis* ssp. *lactis* 32 strain (the only nisin-A producing strain) was obtained from Laval university culture collection. All LAB strains were activated for 16 h in De Man, Rogosa and Sharpe (MRS), Nutri Bact, Canada, broth medium at 37 °C and transferred once before use. The *Clostridium tyrobutyricum* ATCC 25755 was activated in Reinforced Clostridial Medium (RCM), Nutri Bact, Canada, under anaerobic conditions at 37 °C and transferred once [24].

### 2.2. Spore Production

Spores were prepared according to a published method [25] with the following modifications: RCM broth was inoculated with culture (1% *v*/*v*) and incubated for two weeks at 37 °C in an anaerobic chamber (Forma Scientific, Inc., Ohio, OH, USA). The spores were collected by centrifugation at 5000× *g* at 4 °C for 10 min and resuspended in sterilized reconstituted skim milk, Nutri Bact, Canada, (12% *w*/*v*). The spore suspension was heated at 85 °C for 5 min to kill the vegitative cells. Spores were enumerated using RCM agar and incubated under anarobic condition at 37 °C [24].

### 2.3. Pearce Test

To select the best ratio between *Lactococcus* strains the Pearce test was done. The *Lactococcus* strains were grown in MRS medium for 16 h at 30 °C. Cultures were centrifuged at 5000× *g* for 5 min and the pellets were washed twice with 0.85% saline solution. All strains were standardized to an optical density of 0.2 at 600 nm using 0.15% peptone water, Nutri Bact, Canada. Sterilized reconstituted skim milk (40 mL, 12% *w*/*v*) was inoculated with standardized strains suspension at different ratios and incubated on a Cheddar cheese temperature cycle [26,27]. Samples were taken aseptically at 0 h, 2 h and 5 h for the determination of pH, bacterial count and bacteriocin production.

### 2.4. Nisin Encapsulation

Nisin in a commercial form (Nisaplin^®^, 2.5% nisin, Cayman Chemicals, Ann Arbor, MI, USA) was purified on a Sep-Pack C18 column at 4 °C at a flow rate of 3 mL/min and concentrated with Speed-Vac for 6 h at 50 °C Its concentration was measured by High-performance liquid chromatography (HPLC) [24]. It was then microencapsulated in 1% sodium alginate (Sigma Aldrich, Oakville, ON, Canada) with 0.5% non-gelatinized resistant starch (Manitoba Starch products Inc, Carberry, MB, Canada) as described previously [26] using an Inotech IE-20 encapsulator (Inotech Biosystems International, Inc., Rockville, MD, USA) with a 300 µm nozzle and syringe pump set at a flow rate of 2.91 mL/min to form droplets stabilized on contact with sterile 5% calcium chloride solution at pH 6.00. The encapsulator vibration frequency was set at 2000 Hz. The resulting beads were stirred at 300 rpm for 4 h at room temperature. The encapsulation efficiency was determined as described previously. Microcapsules were added to milk to obtain a nisin concentration of 50 µg/mL [24]. 

### 2.5. Cheese Slurry Preparation and Experimental Treatments

The cheese slurry was prepared as described previously [26]. Pasteurized milk was acidified with glucono-δ-lactone (Sigma, St-Louis, MO, USA). The unsalted fresh curd was lyophilized, then reduced to powder in a Quadro ComIL disc grinder (Waterloo, ON, Canada) and irradiated at 5 kGy with Cobalt 60 for 2 h. The pH was adjusted with concentrated lactic acid. The experimental formulations are shown in Table 1.

Each slurry was divided into three groups of 100 g as follows: Group 1: Cheddar cheese starter (8 log_10_ cfu/g) + *C. tyrobutyricum* (4 log_10_ cfu/g)Group 2: Cheddar cheese starter (8 log_10_ cfu/g) + *C. tyrobutyricum* (4 log_10_ cfu/g) + encapsulated nisinGroup 3: Cheddar cheese starter (8 log_10_ cfu/g) + *C. tyrobutyricum* (4 log_10_ cfu/g) + *L. lactis* ssp. *lactis* 32 (nisin A producer)
The slurry was incubated for four weeks at 4 °C or for two weeks at 30 °C to accelerate the reactions to mimic the condition after one year. Chemical composition, bacterial counts and free nisin concentration were measured weekly. Weighed samples (about 2.5 g) were blended with 22.5 mL of peptone water in a stomacher (Seward model 400, Norfolk, UK) and diluted serially. For *C. tyrobutyricum* counts, a portion of each dilution was heated to 85 °C for 10 min before plating (1 mL) using RCM medium at 37 °C under anaerobic condition. All cheese analyses were carried out in triplicate. Fat and moisture contents were measured using the CEM (SMART 6 Moisture/Solids Analyzer).

### 2.6. Nisin Determination by HPLC

Nisin extraction from cheese was performed as described in Jarvis, (2016) with some modifications. Two grams of cheese slurry were added to 8 mL of 0.02 N HCl then homogenized for 1 min at high speed before the slurry was boiled at 98 °C for 5 min. The extract was cooled immediately to 20 °C followed by centrifugation at 10,000× *g*, 4 °C for 30 min. The fat layer was removed, and the supernatant was filtered through a 0.2 μm filter before being analyzed by a reverse-phase HPLC (HPLC Hewlett Packard Series 1100; Agilent technologies Canada Inc., Saint-Laurent, QC, Canada) [24].

### 2.7. Metagenomic Analysis

#### 2.7.1. Sample Preparation

One mL of the cheese suspension was centrifuged at 10,000× *g* for 20 min, and the fat layer was removed before the cell pellet was suspended in 500 mL of a TE 2X solution (20 mM Tris HCl pH 8.0, 2 mM EDTA). 1.25 µL of Propidium monoazide (PMA) solution (Biotium Inc., Hayward, CA, USA) was added to bacterial suspension. The tubes were shaken in the dark for 5 min to allow for maximal PMA contact with DNA, then all tubes were exposed to a PMA lamp apparatus under ice (halogen 500W, Ingenia Biosystems, Barcelona, Spain) for 15 min. Finally, all tubes were recentrifuged at 10,000× *g* for 15 min to recover cell pellets for DNA extraction and stored at −20 °C [28] as described below.

#### 2.7.2. DNA Extraction from Cheese Samples

Total DNA was extracted from cheese samples according to Desfossés-Foucault et al. (2012) using Tissue Kit of Gram-positive bacteria (Qiagen, Mississauga, ON, Canada). all DNA samples were kept at −20 °C until analysis [26].

#### 2.7.3. 16S rRNA Sequencing and Analysis

The diversity of the cheese microbiome was assessed by sequencing the bacterial 16S rRNA gene in the V3-V4 region using the amplification primers 341F (5’ CCTACGGGNGGCWGCAG-3’) and 805R (5’-GACTACHVGGGTATCTAATCC-3’) adapted to incorporate the transposon-based Illumina Nextera adapters (Illumina, San Diego, CA, USA) and a sample barcode sequence allowing multiplexed sequencing. High-throughput sequencing was performed at the Institute for Integrative Systems Biology at Université Laval, QC, Canada on a MiSeq platform using 2× 300 bp paired-end sequencing (Illumina, San Diego, CA, USA). The 16S rRNA gene was profiled using demultiplexed raw data processed with Mothur software (v1.35.1) as described previously [27]. Sequences were aligned using the bacterial reference database SILVA with the align.seqs command. The sequences were clustered into operational taxonomic units (OTUs) using the OPTI parameter. Representative OTU sequences were assigned to taxa based on the Green genes reference database [28].

### 2.8. Statistical Analysis

A split-plot design was applied to determine the effect of time on bacterial growth and the effect of NaCl concentration, pH and time on *Clostridium* and starter strain counts. All experiments were performed three times. Effects were declared significant at *p* ≤ 0.05. Statistical analyses were carried out using the SAS general linear models’ procedure (version 9.1.3, Cary, NC, USA).

## 3. Results

### 3.1. Selection of Mixed Culture Ratio and Production of Encapsulated Nisin

All three *Lactococcus lactis* strains grew well in skim milk under Cheddar cheese temperature cycle conditions. Growth, final pH and bacteriocin production varied between strains (Table 2, Table 3 and Table 4). Strain *L. lactis* ssp. *lactis* CUC-H was the strongest acidifier after 5 h of incubation. Production of nisin-A by *L. lactis* ssp. *lactis* 32 started after 5 h. The most compatible combination was *L. lactis* spp. *lactis* CUC-H, *L. lactis* spp. *cremoris* 222 and *L. lactis* spp. *lactis* 32 at a ratio of 1:1:1.5 for a total inoculum volume of 2%. This provides optimal growth and maximal acid production.

### 3.2. Chemical and Microbiological Composition of Cheese Slurry

The slurries had similar chemical compositions, which corresponded to the normal composition of Cheddar cheese (Table 5), except for the NaCl concentration. At 2.0% NaCl, no significant difference was found between pH 5.0 and 5.3 on *Lactococcus* counts. Therefore, only the data of pH 5.3 will be presented. At 2% salt concentration, a slight reduction was seen in the control group at 4 °C after four weeks. On the other hand, the presence of encapsulated nisin was associated with an approximately 1 log_10_ reduction until the second week. The total bacterial counts increased again during the third week (*p* ≤ 0.05), likely due to the gradual release of nisin from capsules. In the presence of the nisin-A producer (strain 32), counts appeared stable over time. At 1.3% NaCl, the same trend overall was observed (Figure 1).

At 30 °C, the total bacterial counts in control fell by about 0.3 log_10_, a 1.0 log_10_ reduction (*p* ≤ 0.05) was noted in the encapsulated nisin-A group at 2% salt, while it was 0.6 log_10_ at 1.3% salt, and a slight reduction appeared in the presence of the protective strain, perhaps due to better bacterial growth at this temperature (Figure 2) at 2% salt. The gradual in situ production of nisin during storage was confirmed by quantitative HPLC. These results show that these NaCl concentrations and pH do not likely interfere with starter culture growth.

In the case of *C. tyrobutyricum* counts, significant reductions were noted after the first week in the presence of encapsulated nisin and in the second week in the protective strain treated group at 1.3% and 2% NaCl at 4 °C (Figure 3). In the encapsulated nisintreated group, no spores were detected after the third week (*p* ≤ 0.05) in RCM medium. At 2% NaCl under 4 °C, none were detected after the second week, suggesting synergism between NaCl and nisin-A. We have also observed a 0.5 log_10_ reduction in *C. tyrobutyricum* counts in the protective starter treated group compared to control (*p* ≤ 0.05) during four weeks of incubation under two salt concentrations at 4 °C. This inhibition activity was associated with the gradual in situ production of nisin during storage, as confirmed by nisin quantification using HPLC. Gas production was observed in the control group at 1.3% NaCl at 30 °C (Figure 4). The *C. tyrobutyricum* count decreased by 0.74 log after one week under both salt concentrations, while the count was maintained constantly at 30 °C after first week (Table 6).

### 3.3. In Situ Nisin Production

Nisin production by *L. lactis* ssp *lactis* strain 32 (protective strain) in cheese slurry was not affected by pH (Figure 5, Figure 6 and Figure 7). Production was gradual, reaching 97.54 ± 1 µg/g at 2% NaCl and 80.92 ± 1 µg/g at 1.3% NaCl after four weeks at 4 °C (Figure 5). A higher salt concentration could increase nisin production by putting the bacterial cells under more stress [28]. Somewhat more nisin was found at 30 °C, which could be due to the suitable temperature for bacterial growth to produce bacteriocins (Figure 7). These results showed that *L. lactis* ssp *lactis* strain 32 was able to survive and produce nisin in cheese slurry during storage at both temperatures. We have also shown that encapsulation improves the stability of nisin and controls its release in cheese during storage. At 4 °C and in the presence of 2% salt, the concentration of released nisin increased gradually within 2 weeks then decreased to about 30 µg/g at 4 °C due to the reduction of nisin amount inside capsules. The rise and the fall were both slower with 1.3% salt (Figure 6). At 30 °C, the rise appears to have occurred within 1 week, followed by the same tendency to drop (Figure 8).

### 3.4. Metagenomics Analysis

The impact of the nisin-A-producer strain and encapsulated nisin-A on the composition of Cheddar cheese slurry microbiota during storage was expressed in terms of the relative abundance of various families of bacteria. Figure 9 and Figure 10 show the changes in the diversity of genera over time. In the protective strain treated group (pH 5.3 at 1.3% salt) incubated at 4 °C, the genus *Lactococcus* was dominant at time zero, then increased significantly to 94% of the total bacteria after four weeks of incubation, followed by the encapsulated nisin treated group then the control group. The greatest drop in relative abundance of *Clostridium* at this temperature was observed in the encapsulated nisin treated group, followed by the bacteriocin-producing strain treated group compare to the control group.

After two weeks at 30 °C and 1.3% NaCl, the relative abundance of *Clostridium* spp. in the control group had increased significantly compared to the other groups. The proportion of genus *Lactococcus* had increased slightly and was still dominant, followed by the genus *Pediococcus* in the control and protective strain treated groups compared to the encapsulated nisin treated group. In this case, the genus *Lactococcus* decreased significantly with the increasing abundance of genus *Pediococcus* during the second week at 30 °C and 1.3% NaCl.

During four weeks at 4 °C and 2% salt, the relative abundance of *Clostridium* spp. increased in the slurry adjusted to pH 5.3 and 2% salt. Meanwhile, *Lactococcus* declined significantly during week four at 2% salt in all groups, while the genus *Pediococcus* reached 85%, 64% and 55%, respectively, in the control, encapsulated nisin and nisin producer strain treated groups. At 30 °C and 2% salt, in the control group the *Clostridium* abundance increased during the two weeks of incubation compare to the encapsulated nisin and the nisin producing strain treated groups (*p* < 0.05) due to the presence of nisin-A.

## 4. Discussion

A great interest has developed around the use of lactic acid bacteria (LAB) as protective cultures for preserving food or to prolong their shelf life. The bio-protective activity of LAB is related to their ability to produce substances having biological activities at very low concentrations, including organic acids (acetic, lactic and propionic acid), hydrogen peroxide, diacetyl, acetaldehyde and antimicrobial peptides such as bacteriocins [29,30].

The present study aimed to assess the potential of bioprotective lactic acid cultures as well as their metabolites with antimicrobial activity (nisin) for the control of *Clostridium tyrobutyricum* in Cheddar-type cheese slurries.

The strains *L. lactis* spp. *lactis* CUC-H, 32 and *L. lactis* spp. *cremoris* 222 were selected to be used as a protective starter. To select the suitable ratio between the selected strains, the Pearce test was done. The results showed that the most compatible ratio was *L. lactis* spp. *lactis* CUC-H, *L. lactis* spp. *cremoris* 222 and lactis 32 at a ratio of 1:1:1.5 for a total inoculum volume of 2%. A combination of a nisin-producing *l. lactis* spp. *cremoris* JSI02 and *l. lactis* spp. *lactis* C2 at a total inoculation volume of 6% (4% C2 and 2% JSI02) reportedly reached a pH ≤ 5.4 in 5 h [31]. In addition, the combinations of *l. lactis* spp. *cremoris* and *l. lactis* spp. *lactis* and *l. lactis* spp. *lactis* biovar *diacetylactis* at a ratio 1.5:1.5:1 for a 2% total inoculum reportedly grew and produced nisin Z [21]. In all these cases, bacteriocin was produced during fermentation of milk. It remains to be shown whether bacteriocin production continues during cheese storage too.

The chemical composition of Cheddar cheese slurries and the effect of pH and salt on bacterial growth and bioactive compound production in milk has been studied [17]. Under the two NaCl concentrations, there is no significant effect of pHs on *Lactococcus* counts. The *C. tyrobutyricum* was not detected in the group treated with encapsulated nisin in the presence of 2% salt, from the second week of storage at 4 °C. The addition of the protective ferment inhibited the growth of *C. tyrobutyricuim* from the second week. Similar results have been reported for four clinical isolates of *Clostridium difficile* 2.5 log_10_ cfu/mL reductions from 10^6^ cfu/mL in the presence of 3.2 µg/mL nisin [32]. *Clostridium beijerinckii* at 2.2 × 10^5^ spores/mL reportedly failed to germinate in pasteurized ovine milk inoculated with a bacteriocin-producing strain of *L. lactis* ssp. *lactis*, thus preventing late blowing even after 30 days of incubation at 23 °C [33].

The nisin production from *L. lactis* ssp. *lactis* 32 was followed up during the Cheddar slurries incubation. The production of nisin was increased gradually for all cheese groups. This is in agreement with studies of nisin Z produced by *L. lactis* biovar *diacetylactis* during ripening of Cheddar cheese, in which activity declined from over 300 units per g to less than 30 units in six months at 7 °C [21]. On the other side, the capsules maintain stability and control the nisin release in cheese during storage. The nisin released from capsules decreased after the second week. Similar results have been obtained for liposome-encapsulated nisin Z during a second month of storage at 7 °C [21]. Moreover, the nisin producing *L. lactis* ssp. *lactis* INIA 415 strain delayed the late blowing defect (LBD) by seven days during ripening at 14 °C in the cheese produced from milk contaminated with 3.1 log spores/ mL of *C. tyrobutyricum* CECT 4011 spores [34]. The same inhibitory effect was observed against *C. tyrobutyricum* CECT 4011 using the *L. lactis* IPLA 729 strain that produces nisin-Z in a combination with starter culture during the manufacturing of Vidiago cheese, a semi-hard farmhouse variety, produced in Asturias, Spain. The counts of C. *tyrobutyricum* were reduced by 3 logs_10_ at nisin concentration of 1600 AU / ml compared to an increase in the control group of 3.2 logs_10_ during the 30 days of storage at 12 °C [35].

The microbial ecosystem of Cheddar cheese slurries was tested, at low salt concentration (1.3% NaCl). The relative abundance of *Clostridium* at 4 °C dropped significantly in the encapsulated nisin treated group, followed by the nisin-producing strain treated group compare to the control group. The effect of lowered salt concentration on bacterial growth in ripening cheese has been studied. In soft cheese surface-ripened for 27 days at 12 °C, *Pseudomonas fragi*, there was an increase even at the concentration of 1.8% NaCl, but it was significantly lower than the growth with a concentration of 1.3% [13]. In postaging, *Listeria monocytogenes* contamination of Cheddar cheese, the *L. monocytogenes* count decreased by 0.14 to 1.48 log_10_ at 0.7 and 1.8% salt concentration, respectively, during 90 days at 4 °C [36].

Artisanal cheeses collected from 11 different regions in Brazil revealed the dependence of microbial population dynamics on production conditions and subsequent processing steps [37]. Although LAB were always dominant, thermophile bacteria such as *Streptococcus* were found dominant in cheeses from the North, Cerrado and Serro regions. In cheeses from the Northeast, mesophiles (*Leuconostoc*) were more likely to dominate. Cheeses from the North also contained high counts of *Lactobacillus* fermentum. Salers cheese produced from raw milk by traditional methods on farms showed growing LAB counts on the first day of production, based on single-strand conformation polymorphism analysis, and was dominated by *Lactococcus lactis*, *Streptococcus thermophilus*, *Lactobacillus pentosus* and *Lactobacillus plantarum* after three months of ripening [38].

The metagenomic profile of Gouda cheese containing PMA showed that viable counts of starter culture strains (5 *L. lactis cremoris*, 2 *L. lactis lactis* and 1 *Leuconostoc mesenteroides*) increased until the brining step, at which point the counts dropped by 2–3 orders of magnitude, based on amplified fragment length polymorphism quantified by lineage-specific qPCR because of the differential inactivation rates. During ripening, the relative abundance of two citrate-positive *L. lactis* ssp. *lactis*, a *L. lactis* ssp. *cremoris* and a *Ln*. *mesenteroides* ssp. *cremoris* increased, while other strains from the same genus disappeared due to the effects of salt and low pH [39]. Furthermore, the *Lactococcus lactis* species showed the highest mean DNA coverage during ripening of semi-hard Maasdam-type cheeses after 12 days in a warm room or 37 days in a cold room. Finally, a study of bacterial dominance and richness in Cotija cheese using high-throughput 16S rDNA sequencing showed that the cheeses were dominated by three genera: *Lactobacillus*, *Leuconostoc* and *Weissella*, followed by more than 500 non-dominant genera grouped into 31 phyla of bacteria and archaea [39].

## 5. Conclusions

To study the effectiveness of biological antimicrobial formulas in cheese, three groups of Cheddar cheese slurry were designed: a control group challenged with *Clostridium tyrobutyricum* spores without any protection; a similar group containing a bacteriocin-producing lactic ferment and another similar group containing encapsulated nisin. All groups included slurry made at pH 5.0 or pH 5.3 and with 1.3% or 2% NaCl and stored at 4 °C or 30 °C. *Clostridium tyrobutyricum* was not detected in the 2% NaCl + encapsulated nisin group after two weeks at 4 °C. In the presence of the protective strain and 2% NaCl, *Clostridium* viable spore counts fell by 1 log after one month at 4 °C. Also, the bacteriocin was produced during storage at 4 °C. Using classical enumeration techniques, it was shown that *Lactococcus* was the dominant genus during storage in all treatment cases. However, 16S rRNA sequencing showed dynamics among bacteria of different genera. After four weeks at 4 °C, *Pediococcus* become dominant in the presence of encapsulated nisin or the protective strain and 2% salt. In the encapsulated nisin group, the genus *Clostridium* decreased during storage at 4 °C, but its 16S rRNA could be detected. It was not countable using classical techniques. Finally, the classical and the molecular quantitative methods both showed that the bacteriocin-producing strain and encapsulated nisin are both able to control the growth of *Clostridium* in Cheddar cheese slurries.

## Figures and Tables

**Figure 1 foods-10-00898-f001:**
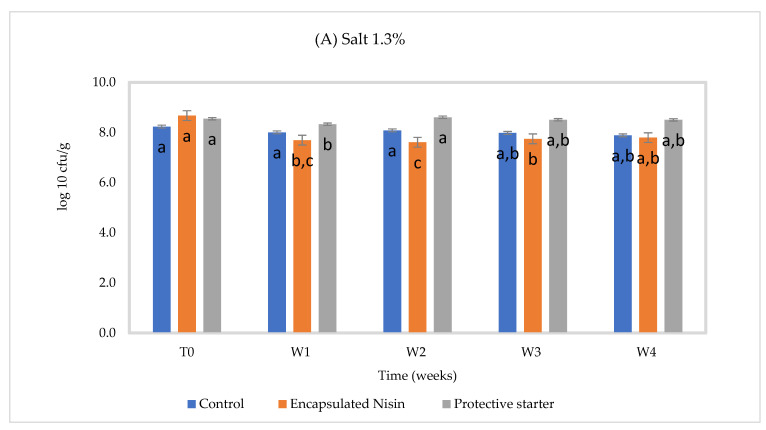
Total lactic acid bacteria count (starter culture) in Cheddar cheese slurries made with (**A**) 1.3% NaCl and (**B**) 2% NaCl, stored for four weeks at 4 °C, pH 5.3 (*n* = 3), different letters indicate significant difference (*p* < 0.05).

**Figure 2 foods-10-00898-f002:**
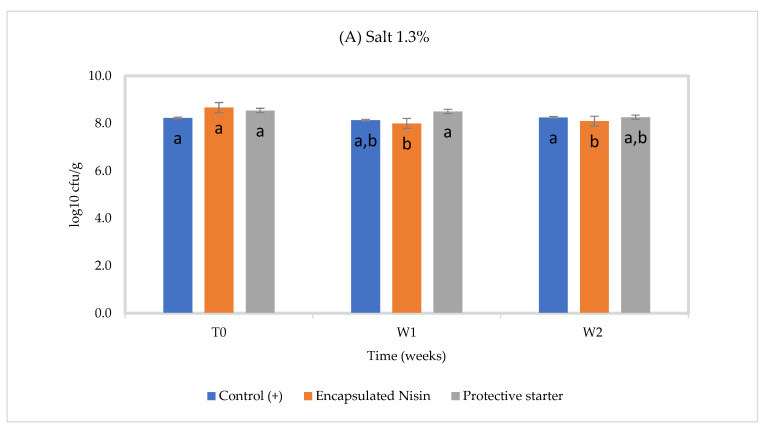
Total lactic acid bacteria count in Cheddar cheese slurries made with (**A**) 1.3% NaCl and (**B**) 2% NaCl stored for two weeks at 30 °C, (*n* = 3), Different letters indicate significant difference (*p* < 0.05).

**Figure 3 foods-10-00898-f003:**
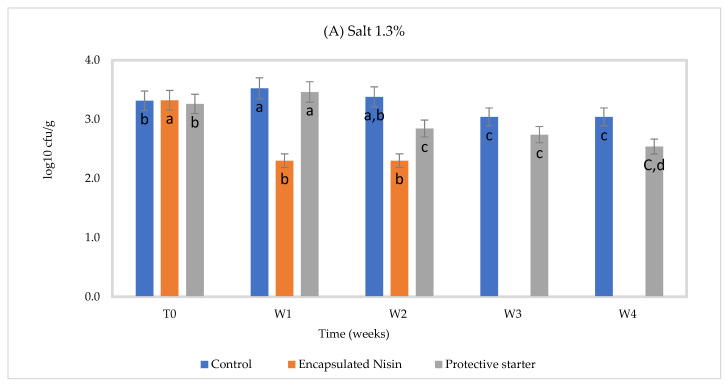
*Clostridium tyrobutyricum* count in Cheddar cheese slurries adjusted to pH 5.3 and stored for four weeks at 4 °C, (**A**) 1.3% NaCl and (**B**) 2% NaCl (*n* = 3), Different letters indicate significant difference (*p* < 0.05).

**Figure 4 foods-10-00898-f004:**
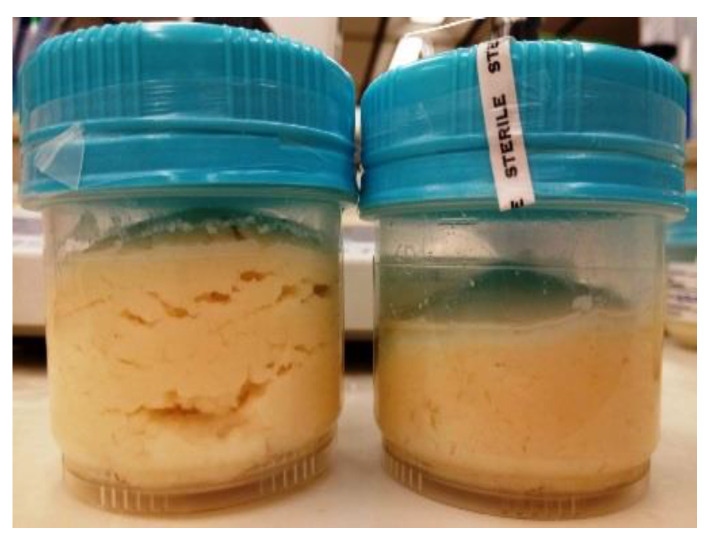
Gas swelling of Cheddar cheese slurry (made with 1.3% NaCl) by *Clostridium tyrobutyricum* after one week at 30 °C, left: control; right: containing encapsulated nisin-A.

**Figure 5 foods-10-00898-f005:**
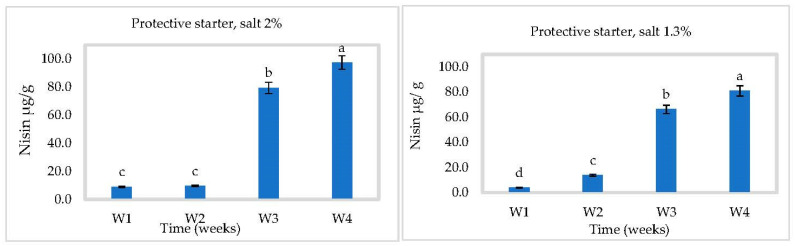
Nisin production in Cheddar cheese slurry containing bacteriocin-producing *Lactococcus lactis* ssp. *lactis* 32 (protective starter), during 4 weeks at 4 °C (*n* = 3), Different letters indicate significant difference (*p* < 0.05).

**Figure 6 foods-10-00898-f006:**
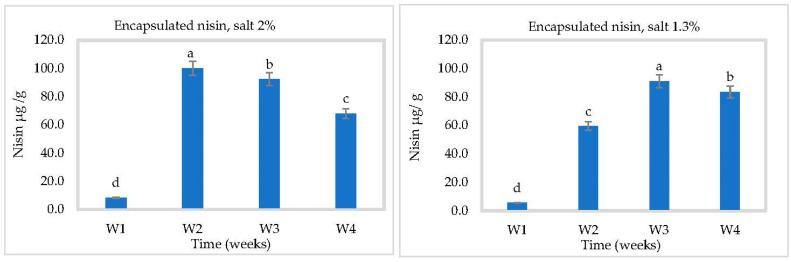
Nisin released in Cheddar cheese slurry containing the encapsulated bacteriocin, during 4 weeks at 4 °C (*n* = 3), W = weeks, Different letters indicate significant difference (*p* < 0.05).

**Figure 7 foods-10-00898-f007:**
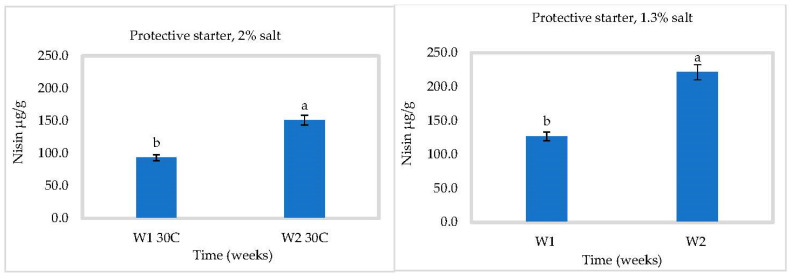
Nisin production in Cheddar cheese slurry containing bacteriocin-producing *Lactococcus lactis* ssp. *lactis* 32 (protective starter), during two weeks at 30 °C (*n* = 3), Different letters indicate significant difference (*p* < 0.05).

**Figure 8 foods-10-00898-f008:**
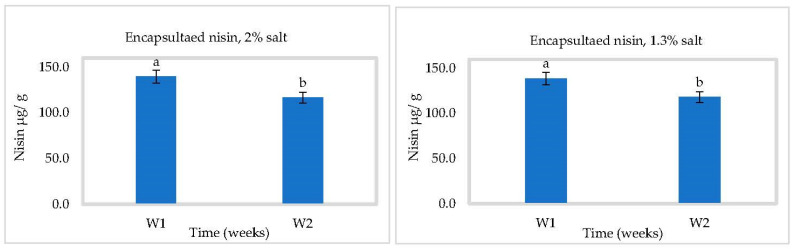
Nisin released in Cheddar cheese slurry containing the encapsulated bacteriocin, during two weeks at 30 °C (*n* = 3), Different letters indicate significant difference (*p* < 0.05).

**Figure 9 foods-10-00898-f009:**
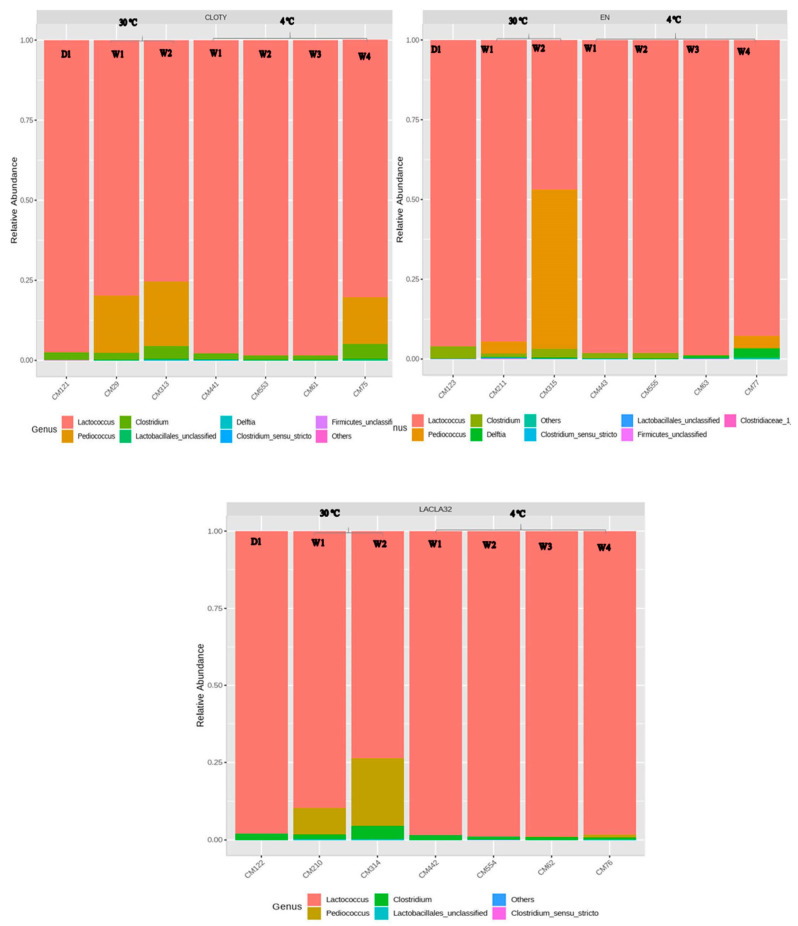
Relative abundance of bacterial genera, based on 16S rRNA, in Cheddar cheese slurries containing *C. tyrobutricum* made with 1.3% NaCl. CLOTY) control, EN) Encapsulated nisin and LACLA32) protective strain treated group, stored for 4 weeks at 4 °C or for two weeks at 30 °C (D1 = time zero).

**Figure 10 foods-10-00898-f010:**
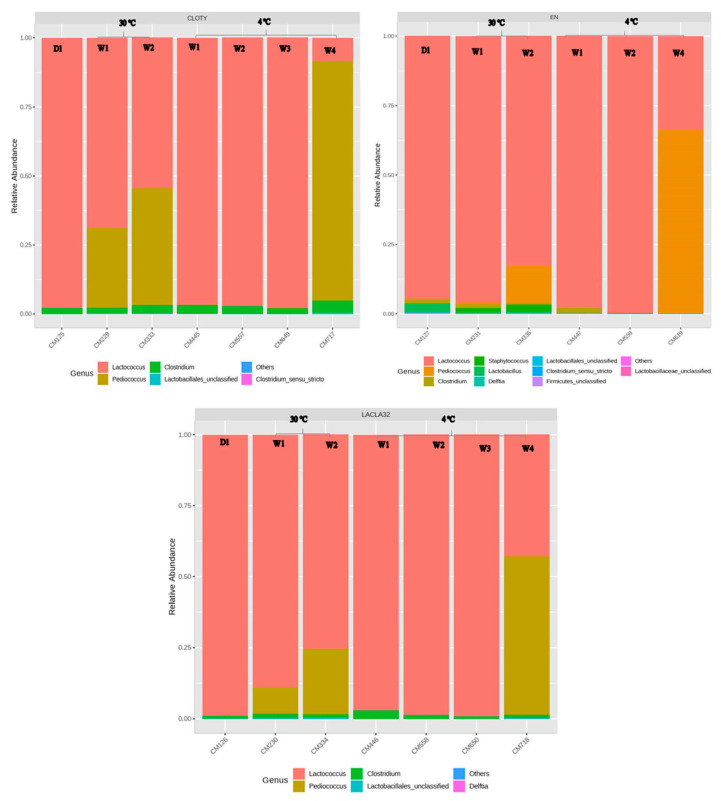
Relative abundance of bacterial genera, based on 16S rRNA, in Cheddar cheese slurries containing *C. tyrobutricum* made with 2% NaCl. CLOTY) control, EN) Encapsulated nisin and LACLA32) protective strain treated group, stored for four weeks at 4 °C or for two weeks at 30 °C (D1 = time zero).

**Table 1 foods-10-00898-t001:** The composition of different Cheddar cheese slurry groups.

Ingredient (g)	pH 5.3		pH 5.00	
	1% NaCl	2% NaCl	1% NaCl	2% NaCl
Cheese powder	180	180	180	180
NaCl solutions pH 2.0	102.9 (NaCl 3.2%)	102.9 (NaCl 3.8%)	102.9 (NaCl 3.2%)	102.9 (NaCl 3.8%)
Starter culture	6	6	6	6
*Clostridium tyrobutyricum*	6	6	6	6
Phosphate buffer	5.1 or 0.6 ^a,b^	5.1 or 0.6 ^a,b^	5.1 or 0.6 ^a,b^	5.1 or 0.6 ^a,b^
Nisin capsules (EN) ^a^	4.5	4.5	4.5	4.5
Protective strain ^b^	4.5	4.5	4.5	4.5
Total weight	300	300	300	300
Target moisture content	38%	38%	38%	38%

^a^ encapsulated nisin group, and ^b^ protective strain group.

**Table 2 foods-10-00898-t002:** Acidification of skim milk by different strains of *Lactococcus lactis* in pure culture or in combination under the Cheddar cheese temperature cycle.

	pH		
	Time Zero	2 h	5 h
*Lactococcus lactis* ssp. *lactis* 32	6.54 d	6.51 b	5.76 b
*Lactococcus lactis* ssp. *cremoris* CUC 222	6.55 c	6.51 b	5.83 a
*Lactococcus lactis* ssp. *lactis* CUC-H	6.57 a	6.52 a	5.53 g
*Lactococcus lactis* ssp*. lactis* 32 + *Lactococcus lactis* ssp. *cremoris* CUC222 (1:1)	6.57 a	6.51 b	5.67 c
*Lactococcus lactis* ssp. *lactis* 32 + *Lactococcus lactis* ssp. *lactis* CUC-H (1:1)	6.56 b	6.5 c	5.48 h
*Lactococcus lactis* ssp. *cremoris* CUC222 + *Lactococcus lactis* ssp. *lactis* CUC-H (1:1)	6.55 c	6.5 c	5.48 h
*Lactococcus lactis* ssp. *lactis* 32 + *Lactococcus lactis* ssp. *cremoris* CUC222 + *Lactococcus lactis* ssp. *lactis* CUC-H (1:1:1)	6.55 c	6.5 c	5.56 e
*Lactococcus lactis* ssp. *lactis* 32 + *Lactococcus lactis* ssp. *cremoris* CUC222 + *Lactococcus lactis* ssp. *lactis* CUC-H (1.5:1:1)	6.55 c	6.49 d	5.55 f
*Lactococcus lactis* ssp. *lactis* 32 + *Lactococcus lactis* ssp. *cremoris* CUC222 + *Lactococcus lactis* ssp. *lactis* CUC-H (1.5:1:1) 2%	6.55 c	6.44 f	5.23 i
*Lactococcus lactis* ssp. *lactis* 32 + *Lactococcus lactis* ssp. *cremoris* CUC222 + *Lactococcus lactis* ssp. *lactis* CUC-H (2:1:1)	6.55 c	6.48 e	5.56 e

Different letters indicate significant difference (*p* < 0.05).

**Table 3 foods-10-00898-t003:** Viable counts of *Lactococcus lactis* in skim milk in pure culture or in combination under the Cheddar cheese temperature cycle (*n* = 3).

	Viable Count Log10 cfu/mL		
	Time Zero	2 h	5 h
*Lactococcus lactis* ssp. *lactis* 32	6.10 ± 0.01 d	7.46 ± 0.02 c	8.69 ± e
*Lactococcus lactis* ssp. *cremoris* CUC 222	5.40 ± 0.02 h	5.70 ± 0.03 f	7.63 ± 0.02 g
*Lactococcus lactis* ssp. *lactis* CUC-H	6.44 ± 0.02 b	7.40 ± 0.01 d	8.92 ± 0.02 a
*Lactococcus lactis* ssp*. lactis* 32 + *Lactococcus lactis* ssp. *cremoris* CUC222 (1:1)	6.00 ± 0.03 e	7.40 ± 0.02 d	8.60 ± 0.02 f
*Lactococcus lactis* ssp. *lactis* 32 + *Lactococcus lactis* ssp. *lactis* CUC-H (1:1)	6.26 ± 0.01 c	7.49 ± 0.01 b,c	8.86 ± b
*Lactococcus lactis* ssp. *cremoris* CUC222 + *Lactococcus lactis* ssp. *lactis* CUC-H (1:1)	5.85 ± 0.02 f	7.21 ± 0.03 e	8.79 ± 0.01 d
*Lactococcus lactis* ssp. *lactis* 32 + *Lactococcus lactis* ssp. *cremoris* CUC222 + *Lactococcus lactis* ssp. *lactis* CUC-H (1:1:1)	5.70 ± 0.03 g	7.51 ± 0.03 b	8.78 ± 0.01 d
*Lactococcus lactis* ssp. *lactis* 32 + *Lactococcus lactis* ssp. *cremoris* CUC222 + *Lactococcus lactis* ssp. *lactis* CUC-H (1.5:1:1)	6.01 ± 0.01 e	7.41 ± 0.01 d	8.83 ± 0.01 c
*Lactococcus lactis* ssp. *lactis* 32 + *Lactococcus lactis* ssp. *cremoris* CUC222 + *Lactococcus lactis* ssp. *lactis* CUC-H (1.5:1:1) 2%	6.24 ± 0.04 c	7.81 ± 0.0 a	8.87 ± 0.01 b
*Lactococcus lactis* ssp. *lactis* 32 + *Lactococcus lactis* ssp. *cremoris* CUC222 + *Lactococcus lactis* ssp. *lactis* CUC-H (2:1:1)	6.60 ± 0.01 a	7.46 ± 0.01c	8.80 ± 0.01 d

Different letters indicate significant difference (*p* < 0.05).

**Table 4 foods-10-00898-t004:** Nisin-A production by the nisin-producing strain of *Lactococcus lactis* in skim milk; pure culture or combination under the Cheddar cheese temperature cycle (*n* = 3).

	Nisin-A (AU/mL)		
	Time Zero	2 h	5 h
*L. lactis* 32	ND	ND	64
*Lactococcus lactis* ssp*. lactis* 32 + *Lactococcus lactis* ssp. *cremoris* CUC222 (1:1)	ND	ND	64
*Lactococcus lactis* ssp. *lactis* 32 + *Lactococcus lactis* ssp. *lactis* CUC-H (1:1)	ND	ND	64
*Lactococcus lactis* ssp. *lactis* 32 + *Lactococcus lactis* ssp. *cremoris* CUC222 + *Lactococcus lactis* ssp. *lactis* CUC-H (1:1:1)	ND	ND	64
*Lactococcus lactis* ssp. *lactis* 32 + *Lactococcus lactis* ssp. *cremoris* CUC222 + *Lactococcus lactis* ssp. *lactis* CUC-H (1.5:1:1)	ND	ND	64
*Lactococcus lactis* ssp. *lactis* 32 + *Lactococcus lactis* ssp. *cremoris* CUC222 + *Lactococcus lactis* ssp. *lactis* CUC-H (1.5:1:1) 2%	ND	ND	256
*Lactococcus lactis* ssp. *lactis* 32 + *Lactococcus lactis* ssp. *cremoris* CUC222 + *Lactococcus lactis* ssp. *lactis* CUC-H (2:1:1)	ND	ND	64

ND: none detected, AU: Arbitrary unit.

**Table 5 foods-10-00898-t005:** Chemical analysis of the cheese slurries at time zero.

pH 5.0 and 1.3% salt				
Groups	NaCl %	Moisture %	Salt/Moisture (S/M)	pH
Control (+)	1.36 ± 0.01	38.49 ± 0.1	3.52 ± 0.03	5.10 ± 0.01
Encapsulated nisin	1.25 ± 0.01	38.53 ± 0.1	2.87 ± 0.01	5.12 ± 0.01
protective strain	1.27 ± 0.01	38.42 ± 0.2	3.32 ± 0.01	5.00 ± 0.01
pH 5.0 and 2% salt				
Groups	NaCl %	Moisture %	(S/M)	pH
Control (+)	2.01 ± 0.01	37.37 ± 0.02	5.38 ± 0.01	5.00 ± 0.02
Encapsulated nisin	1.96 ± 0.01	38.26 ± 0.03	5.12 ± 0.02	5.10 ± 0.01
protective strain	1.86 ± 0.01	37.65 ± 0.02	4.94 ± 0.01	5.00 ± 0.01
pH 5.3 and 1.3% salt				
Groups	NaCl %	Moisture %	(S/M)	pH
Control (+)	1.34 ± 0.01	37.66 ± 0.02	3.56 ± 0.01	5.25 ± 0.01
Encapsulated nisin	1.33 ± 0.01	39.03 ± 0.01	3.41 ± 0.01	5.24 ± 0.01
protective strain	1.23 ± 0.01	38.30 ± 0.02	3.21 ± 0.01	5.24 ± 0.01
pH 5.3 and 2% salt				
Groups	NaCl %	Moisture %	(S/M)	pH
Control (+)	1.88 ± 0.02	37.19 ± 0.03	5.05 ± 0.03	5.26 ± 0.05
Encapsulated nisin	1.87 ± 0.02	37.37 ± 0.02	5.20 ± 0.02	5.28 ± 0.01
protective starter	1.87 ± 0.02	37.38 ± 0.01	5.00 ± 0.01	5.28 ± 0.01

**Table 6 foods-10-00898-t006:** *Clostridium tyrobutyricum* counts in cheese slurry at 30 °C (*n* = 3).

Treatment	Log cfu/g at 2% NaCl		
	Time Zero	Week 1	Week 2
Control	3.53 ± 0.02 b	3.70 ±0.05 a	3.57 ± 0.05 b
Encapsulated nisin	3.54 ± 0.04 a	2.48 ± 0.01 b	2.30 ± 0.02 c
protective starter	3.56 ± 0.02 a	3.51 ± 0.05 a	3.10 ± 0.01 b
Treatment	Log cfu/g at 1.3% NaCl		
	Time Zero	Week 1	Week 2
Control	3.32 ± 0.02 a	3.34 ± 0.02 a	Uncountable (high number)
Encapsulated nisin	3.32 ± 0.02 b	2.54 ± 0.01 c	3.81 ± 0.04 a
protective starter	3.26 ± 0.03 b	3.23 ± 0.03 b	3.80 ± 0.04 a

Different letters indicate significant difference (*p* < 0.05).

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
