# Peer review of "Impact of Nisin and Nisin-Producing Lactococcus lactis ssp. lactis on Clostridium tyrobutyricum and Bacterial Ecosystem of Cheese Matrices"

_foods, 2021, doi:10.3390/foods10040898_

Round 1

Reviewer 1 Report

The main goal of this work was to evaluate the effect of lactic acid cultures and the encapsulated bacteriocin nisin for the control of Clostridium tyrobutyricum in Cheddar-type cheese slurries. Furthermore, two temperatures, two pH values and two NaCl concentrations were used in the evaluation. This is an interesting study that provide some information to the current knowledge.

The article is well written, however there are some corrections to perform such as all bacterial names must be presented in italic (lines 123-127, 179, 518-519, 534), all abbreviations must be described in the text first (see lines 145 and 157) also some wrong points, so a careful read of the manuscript must be performed. Also, i suggest a english text revision.

Specific comments:

  1. In the abstract section it would be important to include a conclusion sentence to give more information to the readers.
  2. In the section introduction there are some errors that need to be addressed. First in line 62 “Two types are known, which differ by a single 62 amino acid at position 27, histidine in nisin-A and asparagine in nisin Z” this is an imprecise sentence because there are several natural forms of nisin and not only two. The sentence must be re-written. Also, in line 66 “It was discovered in 1956…” is not correct, nisin was discovered in 1928 and it has been used in the food industry since 1953, so please verify this information.
  3. In the methods section (2.4 nisin preparation) the authors refer that the microcapsules of nisin were added to milk to obtain a concentration of 2.44 ug/mL, however then in table 1 it is mentioned that 4.5g of encapsulated nisin were added to the moisture. Also, then in the results the encapsulated nisin measured and presented in figure 5 and 6 reached more than 200 ug/g. How do you explain that? What was the total concentration of encapsulated nisin added initially to the cheese? This point must me clarify.
  4. In the results section several points must be addressed:
    • The title of table 2 is not correct. A total inoculum volume of 2% was also used and it is presented by the authors in the table and results, so the title must include this information. Also, why a total inoculum volume of 2% was not tested with the other bacterial combinations? And why only the proportion of L.lactis CUC-H was changed?
    • Table 4 must include a legend with the meaning of ND.
    • All figures do not present the meaning of the letters a,b and c. correct that.
    • It is necessary to include information regarding the Clostridium tyrobutyricum counts in Cheddar cheese stored for four 2 weeks at 30°C, a new figure must be included.
    • The description of Figure 6 is wrong, it is not the evaluation during 4 weeks at 4°C, but yes 2 weeks at 30°C.
    • In lines 379-381 the authors describe that the decrease in Lactococci during the second week could be due to the activity of nisin A. This is really the most probably conclusion? Did nisin usually have antimicrobial activity against its producers??? This sentence must be clarified.
  5. In the Discussion section the authors describe in lines 482-483 that one of the goals was the prevention of texture and flavour side effects, however I think that this goal was not achieved, so some additional information regarding this issue must be addressed or integrated. In fact, in this section from line 523 to 546 there are a lot of information about other cheeses microbiota but without any integration with the experimental work performed. For this reason, it is important to integrate all this information with the work performed. I suggest rewriting these paragraphs.
    • Also, in this section lines 499-500, the authors refer “The addition of the protective ferment inhibited the growth of tyrobutyricuim from the third week only”, however in the figure 3 we can see a reduction of the bacterial count after the week 2 with 1.3% salt. It is not significant this decrease at week 2 ?. The same is described in the results section (lines 290 to 298) and show be review. Also, it is possible to see a high decrease in bacterial counts in week 1 for encapsulated nisin also, that is not mentioned.

Reviewer 2 Report

Impact of Nisin and Nisin-Producing L. Lactis Ssp. Lactis On Clostridium Tyrobutyricum and Bacterial Ecosystem of Cheese Matrices

The topic is of interest.

I am not sure whether the title is chosen correctly. The impact on the bacterial ecosystem based on defined culture (one strain Clostridium tyrobutyricum, one strain Lactococcus lactis, one strain Pediococcus acidilactici, one strain Lactococcus lactis ssp. cremoris) besides nisin and
nisin producing strain (belonging to the flora), was the different pH value and the two NaCl concentrations. The base of the cheese matrix remained the same.

The manuscript contains an immense number of flaws, it should have been through a quality check before submission.

The introduction is easy to read and gives the reader a good and brief overview. But the manuscript process presupposes knowledge that the reader cannot have at that moment. The authors go into the details of the investigation before giving an overview of the entire experimental design. The reader has to do the puzzle.
There are many ambiguities and the presentation of results needs to be improved.

In detail:
Please write out the abbreviations in the title. Do not use capital letters for the species.
l. 15 Full stop after spp is missing.
l.23 Lactococcus should be written out in italic letters.
l. 24 Pediococcus should be written out in italic letters.
l. 32 Clostridium (C.) butyricum
l. 37 C. butyricum is not the only one Clostridium which is responsible for late blowing.
l. 39 Which kind of “very efficient” methods do you mean? Which of them are causing significant modification of milk composition?
l. 56 LAB- please write out once
l. 60 “Bactibase reference”?
l. 62 by strains of which Lactococcus species?
l. 66 Nisin was discovered…

Material and Methods

An introductory sentence on the overview of the methodological approach is missing. E.g. what is the bacterial preparation for? Why are you preparing a nisin-sensitive strain? What is a Pearce
test? Why are you applying this test? …
l. 81: All LAB in one broth?
l. 89: By heating to 85°C for 5 min the vegetative Clostridia are killed. The spores remain (useful for spore enumeration by culturing). You cannot induce germination by this method.
And why do you want to induce germination?
l. 94: Standardized to an optical density- in which medium?
l. 99: It is more about nisin encapsulation as preparation.
l. 103: Sodium alginate, non-gelatinized starch… please specify the manufacturer. Why did you choose these components?
l. 104: The proportions are not clear to me (1% of what? 0.5% of what?)
l. 104: Please cite correctly.
l. 106: Country is missing.
l. 114: What does whole milk mean? It is not the same as in Europe or Asia…
l. 112: Why have you chosen this method? Why are you lyophilizing and irradiating?

Table 1
Please delete/revise this table.
 If this table deals with slurry groups, it would mean e.g. that there is encapsulated nisin and protective strain mixed together in one slurry.
 Why was the culture added in grams? Cfus are important in this case! Why is it solid, wasn’t it a suspension?
 Where is the target moisture from? Why is it interesting? Where is the method for the moisture determination?
 NaCl is given in %, not in g?
 There is something wrong about the total weight. The total weight is not really important in this case.
The different test mixtures must be presented more clearly!
Basically, you have 3 groups:
1. Slurry with starter + Clostridium
2. Slurry with starter + Clostridium + encapsulated nisin
3. Slurry with starter + Clostridium + nisin producing strain
What about a 4th group without Clostridium?
All of those groups underwent following modifications:
1 % and 2 % NaCl addition,
each of the NaCl variations were adjusted to pH 5.3 and 5.0
each pH and NaCl combination of each of 3 groups was halved, one was incubated at 4°C and the other at 30°C right?
l. 123: Where are the cfus from? Please write “log10” correctly.
Be precise with terms: What is the starter culture? What is the Cheddar cheese starter? CUCH?
CUC 222?
When was the Pediococcus strain added?
l. 128: plural
l. 128: What do you mean by chemical composition?
Justify why incubation was carried out for different periods of time!
l. 129: How were these parameters measured?
l. 131: You did not get the total number of clostridia, it´s only the number of spores.
l. 152: Is the DNeasy Blood and Tissue Kit suitable for DNA extractions from spores? To the best of my knowledge, it is not. That would provide lower total levels of clostridial DNA.
Table 2
Please write out the species in all of the tables.
l. 193: Precise the Cheddar cheese temperature cycle please (Materials and Methods).
l. 194: Significant difference between what?
Tab 4
Define ND (not determined?)
l. 204: “I found” Please rewrite.
Tab. 5
What is Salt /moisture?
Figures:
Fig1 made of two figures, please separate.
Wasn’t it 1 % of NaCl?
Please correct labels (e.g. LAB count instead of bacterial growth)
Avoid repetitions in the captions (e.g. remove 4C,..)
What do the letters a, b, c ... stand for?
l. 256: W=week
l. 337: for discussion
Figure 5 is made of 4 figures, please separate.
Please use same scales in all of those figure 5 figures.
Perhaps it would be better to summarize the numbers for the protective starter in one bar chart and the numbers for the encapsulated nisin in another chart.
You used identical signatures for Fig 5 and Fig 6.
Figure 7 and 8
Relative abundance should be given in %.
Please correct the typos.
The bar charts are not easy read. Please concentrate on Lactococcus, Clostridium and Pediococcus. Summarize the “others”.
Use clear and constant abbreviations throughout the manuscript!
l. 481: Which bacteriocins?
l. 482: …and therefore the prevention of texture …
l. 485-486: This sentence is also needed in the Material and Methods!
l. 560: If you mean the clostridial count by culturing: you killed the vegetative forms by heating.
The DNA remained in the sample. Right?

Nisin seems to work. What about the flavor and the consistence of the cheese?
Was the nisin production of other strains excluded?
How homogeneous were the samples taken? From the middle? From the surface?

Reviewer 3 Report

The authors investigated the efficacy of nisin for controlling C. tyrobutyricum growth in Cheddar cheese, comparing the action of mixture of nisin-producing Lactococcus lactis strains with that of encapsulated nisin, considering different conditions of temperature and salt concentration.

The reported results are very interesting because they have shown the effectiveness in reducing the development of Clostridium spores thanks to encapsulated nisin and to the use of starter cultures able to produce bacteriocin even at low refrigeration temperatures. However, in the case of encapsulated nisin the effect on the reduction of spores is evident, while in the case of the mixed culture the reduction (although significant) may not be sufficient to inhibit gas production in the final products. Moreover, it would be appropriate to consider if the concentration of encapsulated nisin is compatible with the legal limits imposed for the use of this bacteriocin as a food additive in cheeses.

Considering the quality of the presentation I have often found confusing and disordered sentences, especially within “materials and methods” and “results” chapters. In addition, references should be revised, since I have found differences between what is reported by the authors and the cited articles.

Specific comments:

P1: Line 15. “The effectiveness of nisin-producing Lactococcus lactis ssp. lactis as a protective strain for control….” In the study, the efficacy of a mixture of strains was evaluated, including L. lactis subsp. lactis, but also L. lactis subsp. cremoris.

P1: Line 15. “spp.”

P1: Line 18. “…Spores were not detected for third week…” should be “for three weeks.”

P1: Line 22. it would be appropriate to indicate in which week the concentration of 97 μg/g has been reached.

P1: Line 23. “Lactococcus” must be in italics.

P1: Lines 38-42. I would also add that in the case of Protected Designation of Origin (PDO) cheeses these procedures are not allowed. In the case of these products the economic loss due to late blowing is very high.

P2: Line 55. I would avoid writing “pure.”

P2: Line 56. You must first indicate “Lactic Acid Bacteria (LAB)”. Then you can use the abbreviation LAB

P2: Line 60. I do not find BACTIBASE in references.

P2: Line 61. “…nisin remains the only one approved… “ by whom? I suggest referring to the Food and Drug Administration “GRAS” list and to the European Food Safety Authority “QPS” list. Moreover, it would be interesting to add in which categories of products it is authorized and which are the legal limits.

https://www.fda.gov/food/generally-recognized-safe-gras/microorganisms-microbial-derived-ingredients-used-food-partial-list

https://www.efsa.europa.eu/en/efsajournal/pub/5063

P2: Lines 77-83. it would be appropriate to indicate how the strains were chosen, indicating studies that have been taken as reference.

P2: Line 77. you can abbreviate the genus “Lactococcus” and “Clostridum” from this point forward.

P2: Line 81. MRS broth

P2: Line 83. “… in anaerobic conditions …” at what temperature? I suppose 37°C.

P2: Line 90. You have to indicate incubation conditions in RCM agar.

P2: Line 92. MRS medium.

P3: Line 95. “ml” must be written “mL” (as well for the whole text)

P3: Line 104. “Hassan, Gomaa et al., (2019)” replace it with [25]

P3: Line 113. “(Gardner-Fortier, 2013)” replace it with [26]

P3: Lines 123-127. C. tyrobutyricum and L. lactis ssp. lactis must be in italics.

P3: Table 1. what does it mean "6" for C. tyrobutyricum?

P4: Line 132. it would be necessary to indicate how spores have been enumerated.

P4: Line 137. “10000 x g, at 4°C”

P4: Line 154, 157, 159. Parentheses are missing.

P4: Line 158. “during” should be “for”.

P5: Line 179. “Clostridium” must be in italics.

P5: Lines 187-189. It should be written the full name of the strains: “L. lactis ssp. lactis CUC-H” and not just “strain CUC-H.

P5: Line 187. “(cremoris)” is not correct, it should be “(lactis)”

P5: Table 2. It should be written the full name of the strains. Moreover, I do not understand “different letters indicate significant difference” (P < 0.05). To my opinion the use of these letters, also in the successive tables, results confusing.

P6: Table 4. “AU/mL” why was this unit used? Nisin concentration has always been given as “μg/g” throughout the rest of the text.

P6: Line 204 “I” should be “it was found”

P6: Line 205, 206. the sentence is not very clear. Moreover, “Ph” must be written “pH”.

P7: Line 258. Does the 0.3 log10 reduction in the control sample refer to the 2% Nacl concentration? It should be indicated

P8: Lines 290-298. The description of the results should be better organized. it is difficult to follow.

P9: Line 331. it would have been interesting to include a comparison between the control and the sample with the protection cultures.

P10: Line 335. “L. lactis spp. Lactis 32”

P10: Line 340. L. lactis ssp. lactis strain 32”

P15: Line 475. “Great” should be “great”

P15: Line 477. I think “produce” is more appropriate than “secrete”

P15: Line 489. “pH ≤ 5,4”

P15: Line 491. “…total inoculum of 106 cfu /ml…” I do not think that this information is correctly reported. I report here a piece of the cited article: “…a total inoculation level of 2% (vol/vol). This starter mixture was composed of three lactococci, Lactococcus lactis subsp. cremoris KB, Lactococcus lactis subsp. lactis KB, and Lactococcus lactis subsp. lactis biovar diacetylactis UL719, at a ratio of 1.5:1.5:1 (vol/vol/vol)…”.

P15: Lines 501, 502. The concentration of nisin used in the cited paper was 3.2 mg/mL, not 3.5 mg/mL. Moreover, in the cited paper spore concentration is expressed in CFU of spores, not spores/mL: “…For germination experiments, 106 CFU of spores (ten mL) were incubated in one mL of germination solution…”

P15: Lines 515-517. Sentence is confusing.

P15: 518. “Pseudomonas fragi” should be italics.

P15: 519. “… at 2% NaCl…” in the cited article NaCl is 1.8%. Moreover, it would be more appropriate to write that there was still an increase even at the concentration of 1.8% NaCl, but significantly lower than the growth with a concentration of 1.3%

P15: it would be interesting to point out studies related to the efficiency of protective coltures or nisin to inhibit C. tyrobutyricum using in different cheese types.

(like: Rilla, N., Martínez, B., Delgado, T., & Rodríguez, A. (2003). Inhibition of Clostridium tyrobutyricum in Vidiago cheese by Lactococcus lactis ssp. lactis IPLA 729, a nisin Z producer. International Journal of Food Microbiology, 85(1–2), 23–33. https://doi.org/10.1016/S0168-1605(02)00478-6;

Ávila, M., Gómez-Torres, N., Gaya, P., & Garde, S. (2020). Effect of a nisin-producing lactococcal starter on the late blowing defect of cheese caused by Clostridium tyrobutyricum. International Journal of Food Science and Technology, 55(10), 3343–3349. https://doi.org/10.1111/ijfs.14598)

P16: Line 546. Citation should be [39], but it is missing in the references

P18: Line 653. 39 is missing

Round 2

Reviewer 1 Report

The authors addressed most of the questions referred in the first revision. However, there are some minor issues to be corrected.

1 -All bacterial names must be presented in italic (lines 93, 579,581,582 and 643).

2- All abbreviations must be described in the text, lines 81 (MRS), 151 (PMA), 477 (NSLAB). 604 (LBD)

3- review extra spaces on lines 41, 602 and 622.

4 – line 45 “sush” change to such

5 – All material used must be included its commercial origin (MRS, RCM, Skim milk, peptone water).

6 – Nisin can be produced by Lactococcus and other bacteria so, in line 62, change “produced by Lactococcus lactis” to mainly produced by Lactococcus lactis.

7- Lines 220 it seems that the sentence is not finished, please review this line.

8- Lines 220 to 224 it is not clear if you are speaking about 1.3 or 2 % NaCl. Please clarify.

9 – Lines 482 to 485 it is confuse, so it should be review.

Author Response

Dear reviewer,

Reviewer 2 Report

The manuscript has been improved. But I still have some concerns about the methods.

  1. 129 “The slurry was incubated for four weeks at 4 °C or for two weeks at 30 °C to accelerate the reactions to mimic the condition after one year.”

There is no proof for this assumption. Would the incubation really accelerate the reactions?

Samples from the slurry:

Your response: “the samples were taken from each part of slurry, then mixed and specific weight was taken”

Was the slurry portioned for each week of sampling? If you take 2.5 g of a representative sample out of the container, you will bring some air in the slurry. This would have an influence of its bacterial ecosystem.

DNA extraction:

l.156 – 158: Desfossés-Foucault, É ., et al.,2012 (Assessment of probiotic viability during Cheddar cheese manufacture and ripening using propidium monoazide-PCR quantification) did not perform an extraction from spores. They used the liquids from the DNeasy Blood and Tissue kit (Qiagen) and beads-no columns. According to Qiagen, the Blood and Tissue Kit lysis is not applied or tested for extraction of DNA from spores.

Therefore, an unverified method was used. This does not lead to any results that can be used.

In the first version of your manuscript you wrote: “the mixture of enzymatic lysis (20 mM Tris HCl pH 8.0, 2 mM EDTA, 1.2% Tri-154tonX-100, 20 mg/mL lysozyme (Sigma-Aldrich),10 μL/ml of 5 U/mL mutanolysin (Sigma-Aldrich) was added to cheese samples then incubated at 37°C for one hour. 25 ml of proteinase K and 200 mL of AL buffer were added (Blood and Tissue kit (Qiagen) and incubated at 70 °C during 30 min. The suspensions were transferred to 2 ml column (Blood and Tissue kit (Qiagen) for centrifugation at 10,000 g for 10 min at room temperature. The DNAs were recovered by precipitation from the supernatant by adding 200 ml of ice-cold absolute ethanol, all samples were kept at -20 °C until analysis [28].”

The column from this kit is applied to bind DNA, there is no DNA supernatant after centrifugation but a flow-through with trash.

What is a 2 ml column?

I guess you mixed two different protocols. This leaves doubts about the correct implementation of the method.

Author Response

Dear reviewer,

Reviewer 3 Report

The authors have significantly improved the paper, but there are still some small aspects to correct. After that, I think it deserves.

Specific comments:

P2: Lines 38-42. “…such methods…”

P2: Lines 38-42. I would add “products” after POD.

P2: Line 61. I would write “… by European Food Safety Authority (EFSA) and Food and Drug Administration (FDA)…”

P2: Lines 77-83. I would emphasize that Lactococcus lactis ssp. lactis 32 is the only one which produces nisin among them.

P4: Line 134. “RCM agar”. You should indicate how many mL have been inoculated in agar plate

P4: Line 144. Parentheses are missing

P5: Lines 182-185. It should be written the full name of the strains types.

P6: Table 4. you should specify that “AU/mL” means “arbitrary activity units per milliliter”. Also, like I ask before, why was this unit used while the nisin concentration has always been given as “μg/g” throughout the rest of the text?

P7: Line 219. You should separate pH from 5.

P7: Line 219-220. I would fix the English text as follows: “...on Lactococcus counts, therefore only

the data of pH 5.3 will be presented”

P7: Line 220. “A slight reduction in the control group at 4 °C WAS SEEN”

P9: Line 320. “…and in the second week…”

P9: Line 321. it is necessary to indicate in the text that the temperature is 4°C

P9: Lines 329-330. Replace the dot with a comma: “... salt concentrations, while the count maintains…”

P10: Figure 4. Why did you not include a comparison between the control and the sample with the protection cultures?

P15: Line 489. “pH ≤ 5,4”

P17: Line 581: It should be written the full name of the strains types.

P18: Clonclusions. Can this encapsulated nisin actually be used? It would be appropriate to consider if the concentration of encapsulated nisin is compatible with the legal limits imposed for the use of this bacteriocin as a food additive in cheeses.

P20: Line 760. Delete “43.”

Author Response

Dear reviewer,
